# Demographic and Obstetric Factors Affecting Mental Health of Pregnant Women During COVID-19: EPDS Assessment Study

Sri Jahnavi Adusumilli
*Biomedical Engineering & Informatics*
Indiana University
Indianapolis, USA
srijadus@iu.edu

Shruti Chaitanya
*Biomedical Engineering & Informatics*
Indiana University
Indianapolis, USA
svchaita@iu.edu

Nigama Pervala
*Biomedical Engineering & Informatics*
Indiana University
Indianapolis, USA
npervala@iu.edu

Jamesetta Quiqui
*Biomedical Engineering & Informatics*
Indiana University
Indianapolis, USA
jkquiqui@iu.edu

Saptarshi Purkayastha
*Biomedical Engineering & Informatics*
Indiana University
Indianapolis, USA
saptpurk@iu.edu

*Abstract*—This study investigates the relationship between demographic and obstetric factors and maternal mental health during the COVID-19 pandemic. Using data from Canadian pregnant women (April 2020 to April 2021) and advanced machine learning models, we found significant associations between sociodemographic characteristics and prenatal mental health, as measured by EPDS scores. Maternal education, household income, perceived threat levels, and anxiety scores emerged as key predictors of depression risk. XGBoost, with an accuracy of about eighty-nine percent, and Random Forest, with an area under the curve of 0.99 for severe depression, demonstrated strong predictive performance and offering promising tools for early intervention. However, the study's cross-sectional design, geographically limited sample, and selected variables constrain generalizability and causal inference. Despite these limitations, the findings underscore the importance of integrating predictive models into prenatal care and highlight the urgent need for targeted mental health screenings during public health crises.

*Index Terms*—Pregnancy, COVID-19, EPDS, Machine Learning, Maternal Mental Health, XGBoost, Random Forest

## I. INTRODUCTION

The Coronavirus Disease 2019 (COVID-19) pandemic has precipitated unprecedented global disruption, with particularly profound implications for maternal mental health. Pregnant women represent a uniquely vulnerable population facing compounded stressors during this crisis, including healthcare access limitations, isolation from support networks, and heightened concerns regarding viral transmission [1]. The incidence of psychiatric symptomatology, particularly depression, has demonstrably increased among expectant mothers due to multifactorial determinants encompassing socioeconomic instability, educational attainment, and employment status [2].

Extensive pre-pandemic literature has established pregnancy as a period of elevated psychological vulnerability. Biaggi et al. (2020) identified that approximately 15-25% of women experience significant anxiety or depressive symptoms during pregnancy [3], while Dadi et al. (2020) documented associations between prenatal depression and adverse birth outcomes, including preterm delivery and low birth weight [4]. The pandemic has exacerbated these concerns substantially. According to cross-national epidemiological investigations, findings reveal alarming statistics: over 70% of pregnant women report clinically significant anxiety or grief, with more than 40% meeting diagnostic thresholds for post-traumatic stress disorder, rates substantially higher than pre-pandemic baselines [5].

The Edinburgh Postnatal Depression Scale (EPDS) serves as the gold standard instrument for quantifying depression severity throughout gestational and postpartum periods. However, pandemic-related disruptions have substantially reduced screening opportunities due to decreased healthcare utilization, as pregnant women avoid medical facilities due to infection concerns and healthcare institutions implement restrictions on in-person consultations [7]. This screening deficit creates a critical gap in early intervention opportunities, particularly concerning given that pregnancy-associated depression correlates with dysregulated cortisol production that may adversely affect fetal neurodevelopment [12].

Research during the pandemic has documented a pronounced surge in anxiety and depression prevalence among expectant mothers, with rates ranging from 58% to 72% substantially exceeding pre-pandemic estimates of 10-15% [6]. This concerning epidemiological shift necessitates enhanced surveillance and intervention strategies. Zhou et al. (2024)

reported a substantial increase in maternal mental health concerns, with anxiety rates rising from 19.2% pre-COVID to 53.2% post-COVID, and depression increasing from 17.5% to 33.6%. He also noted that pregnant women experienced unique stressors including fear of transmitting COVID-19 to their offspring [8]. Tauqeer et al. (2023), in a European cross-sectional study, emphasized that women with pre-existing mental health conditions, limited social support, and financial precarity face disproportionate risk, yet systematic approaches to identifying these high-risk subpopulations remain underdeveloped [9].

The present investigation addresses this critical knowledge gap by exploring the complex interrelationship between maternal and demographic characteristics and depression risk during pregnancy, with particular emphasis on identifying key predictive factors through statistical machine learning methodologies. We specifically focus on quantifying the predictive utility of the EPDS within the pandemic context and developing algorithmic approaches to risk stratification that may facilitate targeted intervention deployment. This research has significant implications for clinical practice, potentially enabling healthcare providers to implement evidence-based screening protocols and preventive measures during this ongoing crisis and future public health emergencies.

## II. METHODOLOGY

### A. Data Source and Study Population

We analyzed data from a prospective cohort study of pregnant women in Canada collected between April 2020 and April 2021 during the COVID-19 pandemic, available from the Pregnancy During the COVID-19 Pandemic (PdP) Study [17]. Participants were recruited through pregnancy organizations, healthcare providers, and social media platforms. Eligibility criteria included: age $\geq$18 years, current pregnancy with gestational age $\leq$35 weeks, residence in Canada, and proficiency in English or French. After excluding invalid or incomplete responses, the final analytical sample comprised 10,772 participants.

### B. Variables and Measurements

The primary outcome was depression severity, measured using the Edinburgh Postnatal Depression Scale (EPDS). EPDS scores were categorized according to validated thresholds: none/minimal (0-6), mild (7-13), moderate (14-19), and severe depression (20-30) [11]. The entire variable list can be seen in Fig. 1. Key predictor variables included:

- Demographic factors: maternal age, household income (categorized into 9 levels from <$20,000 to >$200,000), and maternal education (6 levels from less than high school to a doctoral degree)
- Psychological measures: Patient-Reported Outcomes Measurement Information System (PROMIS) Anxiety scores (range: 7-35)
- Pandemic-specific concerns: a perceived threat to unborn baby (range: 0-100) and perceived risk of COVID-19 harm to baby (range: 0-100)

| Variable Name | Description |
|---|---|
| Maternal_Age | Maternal age (years) at intake |
| Household_Income | In 2019, what was the combined income, prior to taxes and deductions, for all household members from various sources. |
| Maternal_Education | less than high school, diploma, high school diploma, college/trade school, undergraduate degree, master's degree, and doctoral degree |
| EPDS | Edinburgh Postnatal Depression Scale from 0 to 30 |
| PROMIS_Anxiety | A scoring system from 7 to 35 measures anxiety severity, where higher scores indicate more severe anxiety. |
| Gestational_Age_At_Birth | Gestational age at birth (in weeks) |
| Delivery_Date | Delivery dates converted to month and year of birth. |
| Birth_Length | Birth length in centimeters |
| Birth_Weight | Birth weight in grams |
| Delivery_Mode | Mode of delivery categorized as either vaginally or by Caesarean-section (c-section) |
| NICU_Stay | Indicates whether the infant was admitted to the Neonatal Intensive Care Unit (NICU) (Yes or No) |
| Language | Language in which the survey was conducted (English or French) |
| Threaten_Life | Perceived threat level to the maternal life during the COVID-19 pandemic, rated on a scale from 0 to 100 |
| Threaten_Baby_Danger | Perceived threat level to the unborn baby's life during the COVID-19 pandemic, rated on a scale from 0 to 100 |
| Threaten_Baby_Harm | Level of worry regarding exposure to the COVID-19 virus harming the unborn baby, rated on a scale from 0 to 100 |

Fig. 1. Description of each variable in the dataset.

Our seven variables were selected based on established literature on perinatal depression and maternal mental health during the COVID-19 pandemic [2], [10], [11]. Below is a brief justification for each variable:

- Maternal Age: Maternal age has been linked to mental health risk during pregnancy. Sayahi et al. (2023) found that older women showed higher vulnerability to anxiety, insomnia, and social dysfunction (R = 0.223, ($p < 0.01$) [2]. Prior studies also show that both younger and older maternal age can be associated with increased depression risk.
- Household Income: Household income is an important factor in understanding financial stress during the pandemic. Cameron et al. (2020) found that mothers with lower income were more likely to report symptoms of depression [10]. This shows that income can affect maternal mental health and should be included as a key variable.
- Maternal Education: Lower educational attainment is often associated with a higher risk of depression. Sayahi et al. reported that education level significantly influenced social dysfunction during the pandemic, supporting its inclusion as a relevant socioeconomic variable [2].
- PROMIS Anxiety: Anxiety is a well-known predictor of prenatal depression. Sayahi et al. found anxiety to be one of the most common mental health symptoms among pregnant women during COVID-19, highlighting its importance as a psychological measure [2].
- Threaten Baby Danger: This variable assesses perceived general danger to the baby during COVID-19. Sayahi et al. (2023) noted heightened fears regarding infant safety during the pandemic, contributing to increased maternal psychological distress [2].
- Threaten Baby Harm: This variable captures specific concerns about direct harm to the baby due to COVID-19. Sayahi et al. (2023) highlighted that fears related

| | Categorical | | Quantitative | |
|---|---|---|---|---|
| | Nominal | Ordinal | Discrete | Continuous |
| Independent Variables | - | Maternal_Education, Household_Income | - | Maternal age, threaten baby danger, threaten baby harm, PROMIS_Anxiety |
| Dependent variable | - | - | - | (EPDS)Edinburg Postnatal Depression Scale (later converted to categorical) |

Fig. 2. Classification of variables based on type and dependency.

- to infection or complications were key contributors to maternal anxiety [2].
- EPDS: The EPDS was selected because it is a well-established, validated measure for assessing depression symptoms during and after pregnancy. Its ability to capture varying levels of symptom severity makes it especially valuable for identifying women at risk and monitoring mental health outcomes in the perinatal period [11].

### C. Data Processing Pipeline

Data extraction utilized Python- Structured Query Language (SQL) integration in Jupyter Notebook. Our preprocessing workflow included several sequential steps: (1) removal of duplicate entries and header artifacts, (2) selection of seven clinically relevant variables based on perinatal depression literature, (3) handling of missing values through complete case analysis rather than imputation to preserve data integrity, and (4) conversion of categorical variables into appropriate numerical formats.

### D. Data Description and Classification

We described the data to offer an understanding of the essence of the data and provide any necessary details required for the analysis or interpretation (see Fig. 1).

We categorized the data into categorical or quantitative and further into independent or dependent variables. We classified maternal education and household income as categorical independent variables, while the remaining variables were classified as quantitative, with EPDS as the dependent variable (see Fig. 2).

### E. Data Preprocessing and Feature Engineering

Our preprocessing pipeline implemented state-of-the-art techniques for clinical data preparation. After removing header artifacts from the dataset, we selected seven clinically relevant variables based on established literature on perinatal depression risk factors. The dependent variable was the Edinburgh Postnatal Depression Scale (EPDS) score, while six independent variables captured demographic and psychosocial dimensions.

Missing data were handled using complete case analysis rather than imputation to maintain data integrity, following recommendations from recent biostatistical literature on handling missingness in psychiatric datasets [13]. We confirmed data type consistency and converted numerical variables to appropriate float representations for mathematical operations. Ordinal variables underwent structured encoding: household

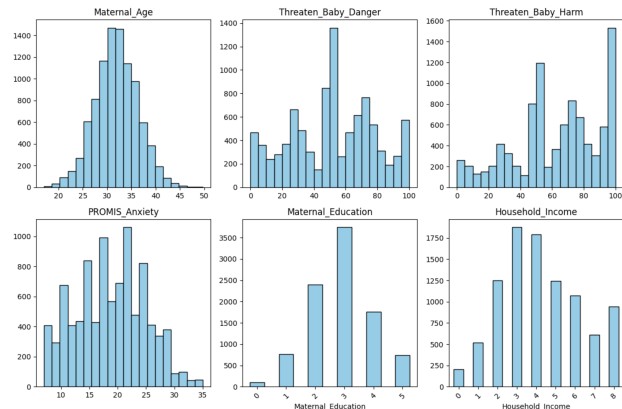

Fig. 3. Histogram representation of each independent variable in the data frame.

income was transformed into a 9-point scale (0-8) representing income brackets from <\$20,000 to >\$200,000; maternal education was encoded on a 6-point scale (0-5) from less than high school to doctoral degree. Following established clinical guidelines [11], we categorized EPDS scores into clinically meaningful groups: none/minimal (0-6), mild (7-13), moderate (14-19), and severe depression (20-30).

For outlier detection, we employed robust statistical methods using boxplot visualization and interquartile range (IQR) analysis with a threshold of $1.5 \times$ IQR, identifying multivariate outliers across psychological measures. We applied mean substitution for outliers, maintaining distribution characteristics while reducing extreme value influence. Exploratory analysis utilized advanced visualization techniques implemented in Python's scientific ecosystem. Descriptive statistics characterized central tendency and dispersion measures across all variables. Distributional analyses through histograms and kernel density estimation revealed patterns in maternal age, anxiety scores, and perceived threat variables. Correlation analysis identified multicollinearity between predictors, informing feature selection decisions for subsequent modeling phases.

We utilized Matplotlib and Seaborn to craft diverse bar charts, histograms, and a correlation heatmap, revealing insights into various variables within our dataset.

### F. Histograms of each Independent variable

We have plotted histograms of each variable (see Fig. 3) to show the frequency distributions within our dataset, providing a clear overview of the data.

## III. STATISTICAL ANALYSIS FRAMEWORK

### A. Distribution Assessment

Prior to inferential analysis, distributional characteristics of all variables were evaluated using Shapiro-Wilk tests and skewness analysis. This assessment determined the appropriateness of parametric versus non-parametric statistical approaches.

## B. Exploratory Analysis

Exploratory data analysis employed descriptive statistics and visualization techniques including histograms, pie charts, scatter plots, and correlation heatmaps to identify potential relationships between variables. Correlation matrix analysis was conducted to assess multicollinearity among predictors, informing subsequent feature selection decisions.

## C. Inferential Statistics

Based on normality testing results, a non-parametric statistical framework was implemented. Chi-square tests with Monte Carlo simulation evaluated associations between categorical variables (EPDS categories with household income and education). Mann-Whitney U tests assessed relationships between ordinal/continuous variables (maternal age, anxiety scores, perceived threats) and depression severity.

Prior to conducting inferential analyses, we rigorously evaluated the distributional characteristics of all variables to ensure appropriate statistical test selection. The Shapiro-Wilk test, selected for its superior power in detecting departures from normality compared to alternative tests such as Kolmogorov-Smirnov, was applied to each continuous variable. All tests yielded p-values below the conventional threshold ($p < 0.05$), providing robust evidence against the null hypothesis of normal distribution.

To further characterize the non-normality patterns, we implemented skewness analysis of all predictor variables. Maternal age, PROMIS anxiety scores, and perceived threat to unborn baby (Threaten_Baby_Danger) demonstrated approximately symmetrical distributions with skewness coefficients between -0.5 and 0.5.

In contrast, the Threaten_Baby_Harm variable exhibited substantial negative skewness (coefficient < -1.0), indicating a pronounced left-tailed distribution with most responses clustered at higher values. This severe asymmetry presented potential challenges for statistical modeling, leading to its exclusion from subsequent analyses in accordance with established statistical practice for handling severely skewed predictors.

## D. Variable Interdependence

The heatmap illustrates the strength and direction of linear relationships among predictor variables, with red indicating positive correlations and blue indicating negative ones. Multicollinearity assessment through correlation matrix analysis (see Fig. 4) revealed a strong positive correlation (r=0.74) between perceived threat variables (baby danger and baby harm), suggesting these measures captured overlapping psychological constructs. Maternal age demonstrated weak negative correlations with both threat perception variables and anxiety (r≈-0.08), while anxiety showed moderate positive correlations with threat perception (r=0.35-0.37). Based on these interdependence patterns and following established feature selection principles [16], we retained only the most discriminative variables for subsequent predictive modeling.

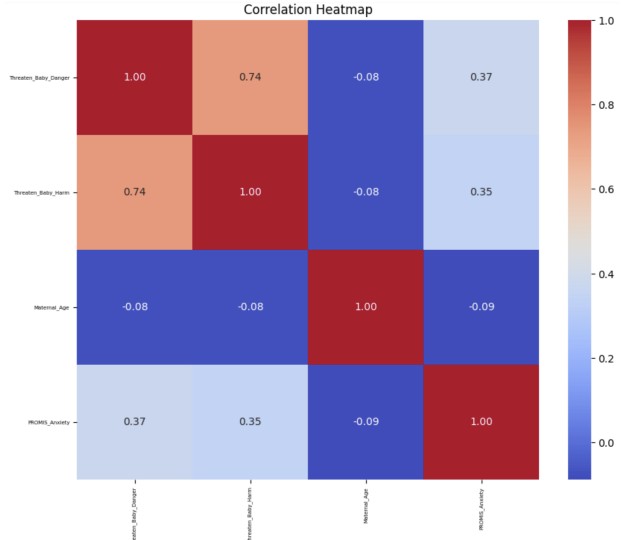

Fig. 4. Correlation heat map among the independent variables.

## E. Machine Learning Framework

We implemented a comprehensive machine learning framework to develop predictive models for depression risk. Features were standardized using RobustScaler to ensure comparable scaling across variables. The dataset was partitioned into training (80%) and testing (20%) sets with stratification to maintain class distribution. We evaluated multiple classification algorithms:

- Logistic Regression
- Random Forest Classification
- XGBoost Classification
- K-Nearest Neighbors

To address class imbalance, we employed the Synthetic Minority Over-sampling Technique (SMOTE) to create balanced training data and enhance the prediction of minority classes. A 5-fold cross-validation approach was used for all models to ensure reliable evaluation of their performance and generalization. Model performance was evaluated using 5-fold cross-validation, accuracy metrics, and Area Under the Receiver Operating Characteristic Curve (AUC-ROC) analysis to assess discrimination capability across depression severity categories.

Feature importance analysis was conducted using ensemble methods to identify the most influential predictors for depression risk, providing insights into potential intervention targets for clinical application. The feature importance analysis identified Maternal Age, Threaten Baby Danger, Household Income, and Maternal Education as the top predictors of depression risk.

## IV. RESULTS

### A. Demographic and Clinical Characteristics

The final analytical sample comprised 10,772 pregnant women recruited during the COVID-19 pandemic in Canada

(April 2020-April 2021). Participants had a mean age of 32.4 years (SD=4.7), with the highest concentration in the 30-35 year range (43.2%). Socioeconomic analysis revealed that most participants reported household incomes in the $70,000-$99,999 (26.2%) and $100,000-$124,999 (24.3%) categories, with fewer respondents in the lowest income brackets (<$20,000: 4.8%; $20,000-$39,999: 5.2%). Educational attainment was high, with undergraduate degrees being most common (37.2%), followed by college/trade school qualifications (22.1%) and master's degrees (16.8%).

### B. Prevalence and Distribution of Depression Symptoms

Analysis of EPDS scores revealed substantial depression symptomatology in the sample. Mild depression was most prevalent (44.8%, n=4,826), followed by none/minimal depression (27.7%, n=2,984), moderate depression (21.8%, n=2,348), and severe depression (5.6%, n=614). This distribution indicates that approximately 72.2% of pregnant women experienced some degree of depression symptoms during the pandemic, with 27.4% reporting moderate-to-severe symptoms, rates substantially higher than pre-pandemic prevalence estimates of 10-15% in comparable populations [14], [15].

### C. Psychosocial Risk Factors

PROMIS Anxiety scores demonstrated elevated levels throughout the sample (mean=19.3, SD=6.2), with most participants scoring in the moderate range (15-25). Perceived threat measures showed distinct patterns: concern about COVID-19 danger to unborn babies exhibited a multimodal distribution with highest frequencies at values 50-55 (mean=51.8, SD=24.6), while perceived harm to babies was heavily skewed toward maximum values (mean=68.7, SD=27.4). This is similar to what has been seen in multi-country findings [5].

Chi-square tests revealed significant associations between EPDS categories and both household income ($\chi^2$=189.46, ($p < 0.001$) and maternal education ($\chi^2$=152.73, ($p < 0.001$). Cross-tabulation analysis demonstrated that higher socioeconomic indicators were protective against depression severity. Among participants with household incomes >$150,000, 42.3% reported no/minimal depression and only 2.9% reported severe depression. Conversely, in the <$40,000 income groups, only 15.6% reported no/minimal depression while 12.7% experienced severe symptoms.

Mann-Whitney tests identified significant associations between depression severity and maternal age (U=3427810, p<0.001), anxiety scores (U=2156489, p<0.001), and perceived threat to baby (U=2876503, p<0.001). Notably, PROMIS Anxiety showed the strongest relationship with EPDS scores (Spearman's $\rho$=0.63), suggesting substantial overlap between anxiety and depression constructs during the pandemic.

### D. Multivariate Correlation Analysis

Correlation matrix analysis revealed important variable relationships. A strong positive correlation (r=0.74) existed

| Models and Accuracies | Logistic Regression | Random Forest | XGBoost | K Nearest Neighbors |
|---|---|---|---|---|
| Accuracy | 65.35% | 63.14% | 62.72% | 60.46% |
| Accuracy after 5-fold cross-validation | 65.40% | 62.77% | 62.72% | 71.92% |
| Accuracy after SMOTE | 64.90% | 77.62% | 89.46% | 65.59% |

Fig. 5. Comparison of model accuracies.

between the two threat perception variables, indicating conceptual overlap. Maternal age showed weak negative correlations with both threat perceptions (r≈-0.08) and anxiety scores (r=-0.09), suggesting slightly lower psychological distress in older mothers. Anxiety demonstrated moderate positive correlations with both threat perception measures (r=0.35-0.37), reflecting the interconnected nature of pandemic-related concerns.

### E. Predictive Modeling Performance

Multiple machine learning algorithms were evaluated to identify optimal approaches for depression risk prediction. While the accuracy of most models remained largely unchanged, the K-Nearest Neighbors (KNN) model showed a notable improvement in accuracy after applying 5-fold cross-validation. After applying SMOTE to address class imbalance, XGBoost achieved the highest accuracy (89.46%), substantially outperforming Logistic Regression (64.90%), Random Forest (77.62%), and K-Nearest Neighbors (65.59%), as can be seen in Fig. 5.

The discriminative ability of these models was assessed through ROC curve analysis (see Fig. 6) for each depression category. The ROC curves assess the classification performance of various models. Curves closer to the top-left corner reflect better predictive performance. For severe depression (Class 3), Random Forest demonstrated exceptional discrimination (AUC=0.99), marginally outperforming XGBoost (AUC=0.98) and KNN (AUC=0.98). Similar patterns emerged for none/minimal depression (Class 0: AUC=0.96, 0.95, and 0.94 for Random Forest, XGBoost, and KNN, respectively) and moderate depression (Class 2: AUC=0.93, 0.90, and 0.91). For mild depression (Class 1), XGBoost exhibited slight superiority (AUC=0.88 vs. 0.87 for Random Forest and 0.81 for KNN).

The key reason for the performance pattern of high AUC but low accuracy was the significant class imbalance in the original dataset. The distribution of depression severity was uneven: 27.7% of participants had no or minimal depression, 44.8% had mild depression (the largest group), 21.8% had moderate depression, and only 5.6% had severe depression, with just 614 cases.

To correct this, we used SMOTE to rebalance the training data, creating 3,429 samples per class and a total of 13,716 balanced samples. The high AUC scores, even with low accuracy, reflect the difference in what these metrics measure. Accuracy looks at exact category predictions, while AUC measures how well the model ranks depression severity across all levels. Our models ranked these levels effectively, which is useful for clinical risk stratification.

Among the models, XGBoost performed best, with sensitivity ranging from 0.84 to 0.97, precision from 0.81 to 0.96,

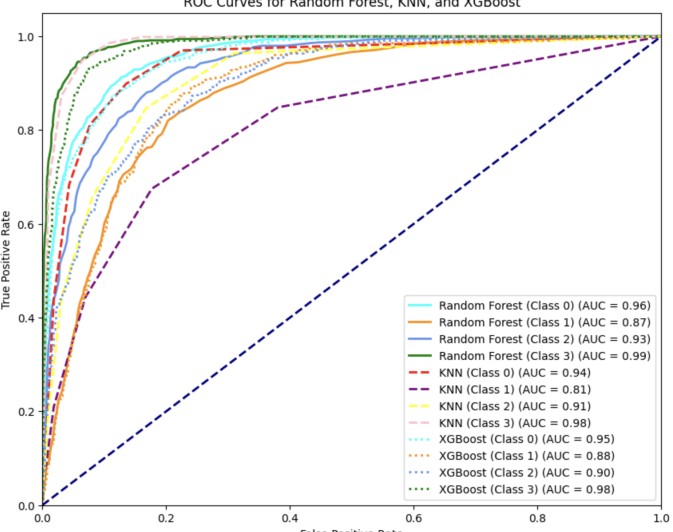

Fig. 6. ROC curves for Random Forest, KNN, and XGBoost models.

```
Cross-validation scores for XGBoost: [0.65962099 0.69814072 0.73423259 0.77834488 0.80021874]
Mean cross-validation score for XGBoost: 0.7341115843243708
Accuracy for XGBoost: 89.46793002915452
              precision    recall  f1-score   support

           0       0.91      0.88      0.89       663
           1       0.81      0.84      0.82       686
           2       0.90      0.89      0.89       694
           3       0.96      0.97      0.97       701

    accuracy                           0.89      2744
   macro avg       0.89      0.89      0.89      2744
weighted avg       0.90      0.89      0.89      2744
```

Fig. 7. Classification report of XGBoost model.

and F1-scores from 0.82 to 0.97 across all classes (see Fig. 7). Random Forest showed exceptional performance at detecting severe depression (AUC = 0.99, precision = 0.89, recall = 0.92) (see Fig. 8). In contrast, KNN performed poorly, with only 39% sensitivity for severe cases.

These findings indicate that the combination of high AUC and low accuracy reflects the challenge of accurately distinguishing between overlapping levels of depression severity. Nevertheless, the model demonstrates strong discriminative ability, which is crucial for clinical screening where prioritizing patients by risk level is more important than exact categorical classification.

### F. Feature Importance Analysis

Random Forest-based feature importance analysis identified PROMIS Anxiety as the most influential predictor of depression risk, accounting for 42.6% of model prediction power. This was followed by perceived threat to baby (23.8%), maternal age (15.3%), household income (10.7%), and maternal education (7.6%). This hierarchy of predictive factors provides crucial insights into the relative contributions of different variables to depression risk during pregnancy in the pandemic context.

These results collectively support the alternative hypothesis that demographic and obstetric factors significantly correlate with depression severity in pregnant women during the

COVID-19 pandemic, with anxiety levels and perceived threat to unborn babies emerging as particularly powerful predictors.

### G. Generalizability of Findings

Although this study was conducted within the Canadian healthcare system, the associations between maternal age, education, income, and mental health are consistent with international findings across varied cultural and healthcare contexts. Sayahi et al. (2023) reported similar associations in Iran, including age-related anxiety and lower education linked to social dysfunction [2]. Phipps et al. (2023) found a 22% depression rate in California, comparable to our 21.8% in the moderate EPDS range [12]. Tauqeer et al. (2023) reported 16–17% depression during Europe's third pandemic wave, aligning with our 27.4% moderate-to-severe rate [9]. Despite cultural and healthcare differences, the consistent impact of demographic and socioeconomic factors suggests broader applicability of our findings, with local adaptation.

## V. LIMITATIONS

Despite the methodological rigor employed, several limitations warrant acknowledgment. The cross-sectional design precludes establishing causal relationships between identified risk factors and depression outcomes. The study's geographical restriction to Canadian participants limits generalizability to other cultural contexts and healthcare systems with different pandemic responses. Self-reported measures, including the EPDS, may introduce response bias, particularly regarding sensitive mental health issues. Additionally, our analysis focused on a predetermined set of demographic and obstetric variables, potentially overlooking other relevant factors such as pre-existing mental health conditions, pregnancy complications, or domestic relationship quality. The convenience sampling strategy, heavily reliant on digital recruitment, may have underrepresented individuals with limited technological access or lower digital literacy, introducing potential selection bias.

## VI. FUTURE WORK

Future research should prioritize several key directions to build upon these findings. Longitudinal studies tracking mental health trajectories throughout pregnancy and into the postpartum period would provide valuable insights into the temporal dynamics of pandemic-related depression. Intervention studies

```
[0.74897494 0.74168565 0.75159526 0.75341841 0.74840474]
Mean cross-validation score for Random Forest: 0.7488158012222191
Accuracy for Random Forest: 77.62390670553935
              precision    recall  f1-score   support

           0       0.80      0.82      0.81       663
           1       0.66      0.62      0.64       686
           2       0.74      0.74      0.74       694
           3       0.89      0.92      0.91       701

    accuracy                           0.78      2744
   macro avg       0.77      0.78      0.77      2744
weighted avg       0.77      0.78      0.77      2744
```

Fig. 8. Classification report of Random Forest model.

testing targeted support strategies for high-risk subgroups identified through our predictive models represent an essential next step toward clinical application. Expanding variable sets to include biological markers, genetic factors, and more comprehensive psychosocial measures would enhance model performance and identify additional intervention targets. Cross-cultural comparisons across diverse healthcare systems and pandemic response contexts would improve generalizability. Assessing model calibration is essential to improve clinical applicability and ensure reliable risk predictions. Finally, implementation science approaches are needed to translate these predictive models into clinically viable screening tools that can be effectively integrated into routine prenatal care, particularly in telehealth contexts that have become increasingly prevalent during the pandemic. Our six-variable predictive model enables real-time risk screening via telehealth, allowing patients to complete brief assessments prior to visits. It is easy to deploy with minimal training, supporting timely care without increasing burden on the clinician.

## VII. CONCLUSION

This study provides compelling evidence that maternal demographic and obstetric factors significantly influence depression risk among pregnant women during the COVID-19 pandemic, supporting our alternative hypothesis. Anxiety levels and perceived threats to unborn babies emerged as particularly powerful predictors of depression severity, followed by socioeconomic indicators, including maternal education and household income. Developing high-performing predictive models, particularly the XGBoost algorithm achieving 89.46% accuracy demonstrates the feasibility of early risk identification based on readily available demographic and psychosocial data. These findings underscore the pandemic's substantial impact on maternal mental health, with 72.2% of participants experiencing some degree of depression symptoms, rates considerably higher than pre-pandemic norms. Our research highlights the urgent need for enhanced mental health screening and targeted interventions for pregnant women during public health crises, with particular attention to those with elevated anxiety, pandemic-specific concerns, and lower socioeconomic resources.

## ACKNOWLEDGMENT

We extend our sincere gratitude to the research team of the Pregnancy During the COVID-19 Pandemic (PdP) Study for providing access to their dataset and the Luddy IUI IT Team for essential computational resources. We are also grateful to the pregnant women who participated in the original study, whose contributions advance our understanding of maternal mental health during public health crises.

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
