# OpenReview forum: "Demographic and Obstetric Factors Affecting Mental Health of Pregnant Women During COVID-19: EPDS Assessment Study"
_IEEE.org/EMBS/BHI/2025/Conference — BHI 2025_

### Official Review · Reviewer_pN2t · 2025-07-13
**Demographic and Obstetric Factors Affecting Mental Health of Pregnant Women During COVID-19: EPDS Assessment Study**

**Confidence:** 5
**Clarity Of Writing:** good
**Clinical Significance:** great
**Methodological Novelty:** great
**Overall Rating:** 7

**Experiments And Results:**

great

**Questions For The Authors:**

None

**Strengths:**

1) In the research in this paper, the main focus was quantifying the predictive utility of the EPDS within the pandemic context and developing algorithmic approaches to risk stratification that may facilitate targeted intervention deployment.
2) This research has significant implications for clinical practice, potentially enabling healthcare providers to implement evidence-based
screening protocols and preventive measures during this ongoing crisis and future public health emergencies.

**Summary Of The Paper:**

The COVID-19 pandemic has precipitated unprecedented global disruption, with particularly profound implications for maternal mental health. The Edinburgh Postnatal Depression Scale (EPDS) serves as the gold standard instrument for quantifying depression severity throughout gestational and postpartum periods.

This study provides compelling evidence that maternal demographic and obstetric factors significantly influence depression risk among pregnant women during the COVID19 pandemic, supporting the researchers' alternative hypothesis.
The research presented in this paper highlights the urgent need for enhanced mental health screening and targeted interventions for pregnant women during public health crises, with particular attention to those with elevated anxiety, pandemic-specific concerns, and lower socioeconomic resources.

**Weaknesses:**

None

---

### Official Review · Reviewer_T2od · 2025-07-16
**Demographic and Obstetric Factors Affecting Mental Health of Pregnant Women During COVID-19: EPDS Assessment Study**

**Confidence:** 4
**Clarity Of Writing:** good
**Clinical Significance:** great
**Methodological Novelty:** fair
**Overall Rating:** 5

**Experiments And Results:**

good

**Questions For The Authors:**

No questions at all. Your study addresses an important clinical and social issue, and the combination of statistical and machine learning methods is a strength. By addressing the points above, you can substantially improve the manuscript’s clarity, rigor, and impact.

**Strengths:**

1. The manuscript addresses an important and underexplored public health issue, focusing on a particularly vulnerable population during the COVID-19 pandemic.

2. The study combines classical statistical analysis with machine learning models to explore predictive factors, reporting high AUC values that indicate good discriminative potential.

3. The clinical implications of risk stratification and targeted intervention are well highlighted.

**Summary Of The Paper:**

The paper investigates depression risk among pregnant women during the COVID-19 pandemic. It combines statistical analysis and machine learning models (e.g., KNN, logistic regression) to identify predictive factors from demographic and obstetric variables, aiming to enhance screening and early intervention. The study reports high AUC values for its models, suggesting good discriminative capability, and discusses potential clinical implications for improving maternal mental health care during public health crises.

**Weaknesses:**

1: Missing citations for variable selection
You mention that seven variables were selected based on prior literature; however, the manuscript does not cite or discuss these studies. Please include these references and briefly describe how each variable is theoretically linked to prenatal depression risk.

2: Modeling claims
While the manuscript describes the use of “sophisticated machine learning algorithms,” the models employed (KNN, logistic regression) are generally considered standard. It would strengthen your work to either:

Use more advanced models (e.g., deep neural networks), or Reframe the description to reflect the models’ actual complexity.

3: Performance paradox: high AUC but low accuracy
This discrepancy requires discussion. Likely causes include class imbalance or dataset size. Please provide details on class distribution and explain why AUC remains high despite low accuracy. Reporting additional metrics such as sensitivity, specificity, precision, recall, and F1-score would help readers better interpret your results.

4: Limited literature coverage
The introduction and discussion could benefit from a broader review of recent studies on maternal mental health during COVID-19 and on machine learning applications in this field. This will better contextualize your contribution.

5: Preprocessing and scaling
The manuscript does not clearly explain which preprocessing steps were performed (e.g., scaling, normalization, encoding) and why they were chosen. Providing this information will improve transparency and reproducibility.
6: Benchmarking:
Your results should be benchmarked with other studies. If this is the first time ML is used on this particular data or study, then it should be mentioned in the claims.

---

### Official Review · Reviewer_uK9y · 2025-07-17
**Demographic and Obstetric Factors Affecting Mental Health of Pregnant Women During COVID-19: EPDS Assessment Study**

**Confidence:** 5
**Clarity Of Writing:** great
**Clinical Significance:** great
**Methodological Novelty:** good
**Overall Rating:** 7

**Experiments And Results:**

good

**Questions For The Authors:**

1. Did you explore calibration of the classification models, especially for clinical applicability in risk stratification?
2. How do you envision integrating the predictive models into real-world clinical workflows, especially for remote or telehealth settings?
3. Were participants asked about prior mental health history? If not, do you anticipate its absence might confound the results?

**Strengths:**

This study is well aligned with BHI themes by integrating machine learning to analyze complex health data from a large cohort. It focuses on maternal mental health during a global health emergency, which is a highly relevant and vital area. The use of a well-validated clinical instrument (EPDS) helps with reliability in outcome measurement. The paper presents a thorough preprocessing and feature engineering pipeline, including robust handling of outliers and missing data. It also applies various supervised learning models, compares their performance, and demonstrates practical application through feature importance analysis. The findings are clinically relevant, especially regarding how anxiety and pandemic-specific concerns influence depression, potentially aiding early intervention strategies in healthcare systems.

**Summary Of The Paper:**

The paper investigates the association between demographic, obstetric, and psychosocial factors and depression severity among pregnant women during the COVID-19 pandemic. The authors utilize data from the Pregnancy During the COVID-19 Pandemic (PdP) Study in Canada, involving 10,772 participants. Depression severity is measured using the Edinburgh Postnatal Depression Scale (EPDS). The paper explores correlations between depression scores and variables including maternal age, education, household income, anxiety levels, and perceived threat to the unborn child. A machine learning pipeline incorporating models such as XGBoost, Random Forest, and K-Nearest Neighbors is used to predict depression risk. Feature importance analysis highlights PROMIS anxiety scores and perceived threats as the most significant predictors. The study also discusses limitations related to self-reported data, cross-sectional design, and sample representativeness.

**Weaknesses:**

Although the dataset is large, generalizability is restricted to Canadian populations with internet access and adequate language proficiency and are not representative of a larger population. The impact on model calibration and real-world applicability is not fully explored in this paper. The explanation of ROC curves and correlation heatmaps is referenced but not detailed in the text, it is important to bring them throughout the results for better accessibility. The paper could benefit from a deeper discussion of the clinical interpretability of the machine learning outputs, especially regarding risk thresholds for intervention. I did not see the abstract in the beginning, make sure to include it in the final draft.

---

### Official Review · Reviewer_KaQy · 2025-07-18
**Using the data samples of 10,272 pregnant women from Canada, this study found that anxiety and perceived threat to the baby were major predictors of prenatal depression during the COVID-19 pandemic. The high symptom rates emphasize the need for stronger mental health care for expectant mothers.**

**Confidence:** 4
**Clarity Of Writing:** excellent
**Clinical Significance:** excellent
**Methodological Novelty:** great
**Overall Rating:** 8

**Experiments And Results:**

great

**Questions For The Authors:**

1. In other contexts, how might the findings apply to non-canadian populations with different healthcare systems or pandemic experiences?
2. Why were variables such as pre-existing mental health conditions or domestic relationship quality excluded?

**Strengths:**

The study’s large sample of 10,772 pregnant women enhances the reliability and statistical power of the findings. The use of multiple machine learning algorithms – Logistic Regression, Random Forest Classification, XGBoost Classification and K-Nearest Neighbors, coupled with SMOTE for class imbalance and 5-fold cross-validation, demonstrates methodological rigor and strengthens predictive modeling for depression risk. The addition of demographic, socioeconomic, and psychosocial factors offers a holistic view of depression risk, supported by clear statistical analyses like Chi-square and Mann-Whitney tests.

**Summary Of The Paper:**

This study explores how demographic and obstetric factors influenced the risk of depression among pregnant women during the COVID-19 pandemic, drawing on data samples of 10,772 participants in Canada. Using the Edinburgh Postnatal Depression Scale (EPDS) to assess symptoms and applying various machine learning models, its identified that XGBoost presented the most accurate predictions. Anxiety levels and perceived threat to the baby emerged as the strongest predictors of depression. Alarmingly, more than 70% of participants reported some level of depressive symptoms, with over a quarter experiencing moderate to severe symptoms-rates significantly higher than pre-pandemic norms. The findings underscore the urgent need for better mental health screening, prenatal care and support for expectant mothers.

**Weaknesses:**

These are already addressed in limitations and future work sections.